# Comparison of the performance of Aptima HIV-1 Quant Dx Assay with Abbott RealTime HIV Assay for viral load monitoring using plasma and Dried Blood Spots collected in Kenya

**Matilu Mwau[1]\*, Sven Schaffer[2]¤, Humphrey Kimani[1], Purity Kasiano[1], Francis Ogolla[1], Elizabeth Ajema[1], Scriven Adoyo[1], Ednah Nyairo[1], Norah Saleri[1], Sangeetha Vijaysri Nair[3]**

1 Kenya Medical Research Institute, Nairobi, Kenya, 2 Hologic, Deutschland GmbH, Wiesbaden, Germany, 3 Hologic, Inc. San Diego, California, United States of America

¤ Current address: Illumina Centre for Molecular Neurobiology Hamburg, Hamburg, Germany
\* mmwau@kemri.go.ke

**Data Availability Statement:** All relevant data are within the manuscript.

## Abstract

### Introduction

HIV-1 viral Load (VL) testing is recommended for the monitoring of antiretroviral treatment. Dried Blood Spots (DBS) are an effective sample type in resource limited settings, where safe phlebotomy and reliable shipping are hard to guarantee. In HIV high burden countries, high throughput assays can improve access to testing services. The Hologic Aptima HIV-1 Quant Dx Assay (Aptima Assay) is a high throughput assay that runs on the CE-IVD approved Panther platform. The objectives of this study were to assess the performance characteristics of Aptima for VL monitoring using plasma and venous DBS specimens and to determine the stability of HIV-1 RNA in DBS.

### Materials and methods

This was a cross-sectional study of 2227 HIV infected adults visiting health facilities in Nairobi and Busia, Kenya. Each provided a venous blood sample; plasma was prepared from 1312 samples while paired DBS samples and plasma were prepared from the remaining 915 samples. The agreement between the Aptima assay and the Abbott RealTime HIV-1 Assay (Abbott RT) was analysed by comparing the HIV-1 VL in both assays at the medical decision point of 1000 copies/mL. To assess stability of HIV-1 RNA in DBS, VL in DBS spotted on day 0 were compared with that from the same DBS card after 21 days of storage at room temperature.

### Results

Overall, 436 plasma samples had quantifiable results in both Aptima and Abbott RT. The agreement between the two assays at 1000 copies/mL was 97.48% with a Pearson's

**Funding:** Hologic, Inc provided the funds and materials for the study.

**Competing interests:** Hologic, Inc. provided the funds and materials for the study. Sangeetha Nair and Sven Schaffer were employees of Hologic, Inc at the time of the study. The rest of the authors are employees of the Kenya Medical Research Institute and declare no conflict of interest. This does not alter our adherence to PLOS ONE policies on sharing data and materials.

correlation coefficient ($r$) of 0.9589 and gave a mean bias of 0.33 log copies/mL on Bland-Altman analysis. For fresh DBS, the agreement in both assays was 94.64% at 1000 copies/mL, with an $r$ of 0.8692 and a mean bias of 0.35 log copies/mL. The overall agreement between DBS tested in Aptima on day 0 versus day 21 was 95.71%, with a mean bias of -0.154.

## Conclusion

The Aptima HIV-1 Quant Dx assay is an accurate test for VL monitoring of HIV-1 using DBS and plasma sample types in Kenya.

## Introduction

More than 70% of the 37.7 million people living with HIV-1 in 2021 were on antiretroviral treatment (ART) [1, 2]. The World Health Organization (WHO) strongly recommends regular viral load (VL) testing to enable early detection of treatment failure, prevent emergence of drug resistance, and minimize transmission [3, 4]. Many countries especially in Africa have scaled up VL testing [5] to meet the WHO recommendations and to help end the AIDS epidemic [2].

In 2020, at least 75% of the 1.4 million people living with HIV-1 in Kenya were on ART and needed routine VL testing to monitor effectiveness of treatment [1]. Through nine reference laboratories and using both plasma and DBS, the national VL program delivered 850000 tests [6]. Most facilities offering comprehensive care for HIV in Kenya are located far from reference laboratories, and therefore DBS collection and transportation enables decentralization of specimen collection and increases access to VL monitoring. It also enables more timely detection for treatment failure which is crucial in reducing HIV transmission and emergence of drug resistance. Approximately 20% of specimens used for VL monitoring in Kenya are DBS.

In Kenya, VL tests are conducted using Abbott and Roche assays running on the m2000 or COBAS Ampliprep/COBAS TaqMan platforms respectively. These platforms are medium throughput and since number of samples often exceeds the capacity to test, backlogs are quite common [7]. These backlogs could be reduced either by building the capacity of regional laboratories to test or by providing platforms with higher throughputs in the reference laboratories.

The Hologic Aptima HIV-1 Quant Dx Assay (Aptima Assay) is an in vitro nucleic acid amplification test that runs on the high throughput Panther Platform. It is intended for use in the detection of Human Immunodeficiency Virus type 1 (HIV-1) in human plasma, serum and DBS from infected individuals [8, 9]. The assay also quantifies HIV in human plasma and DBS over ranges of 30 to 10,000,000 copies/mL and 883 to 10,000,000 copies/mL, respectively. Data on the performance of the Aptima VL Assay has been published mainly from studies conducted in the United States and Europe [10–20]. Few publications on Aptima performance have come out of Africa [17, 19], although Africa has the greatest genetic diversity of HIV-1 [21, 22].

In a previous study [23], we investigated the performance of the Aptima HIV Quant Dx assay using fingerstick and venous dried blood spots prepared under field conditions. In that study, we did not compare HIV-1 VL in Aptima with those in other assays using DBS. Instead it was compared to the Aptima plasma HIV-1 VL. In fact, whereas multiple studies compared VL results of paired DBS and plasma samples at the medical decision point of 1000 copies/ mL by testing both sample types in Aptima [23–26], there is no published information comparing the performance of DBS specimens tested in Aptima and other assays at the medical decision point (MDP) of 1000 copies/mL.

The purpose of this study was to compare the performance of the Aptima assay to Abbott RT assay for VL monitoring using both plasma and DBS specimens collected from patients in Kenya. The stability of HIV in DBS specimens was also assessed in this study.

## Materials and methods

### Ethics statement

This study was approved by the Institutional Review Board of the Kenya Medical Research Institute under protocol numbers KEMRI/SERU 2457 and 3544. It was conducted in accordance with the ethical standards of the Helsinki Declaration of 1975 as revised in 2000.

### Study design

This was a cross-sectional prospective study conducted in 2017 and 2018. Firstly, the performance of Aptima assay on the Panther platform was compared with the reference Abbott-RealTime HIV-1 (Abbott RT) assay using plasma prepared from venous blood samples collected from HIV positive patients enrolled in Nairobi and Busia. The performance of DBS was evaluated from a further 915 venous blood samples collected from HIV positive patients enrolled in Nairobi. All venous blood was sent to the laboratory where it was either separated into plasma or spotted on to DBS cards.

To accurately assess the stability of HIV-1 RNA in DBS, the venous DBS samples prepared for the first comparison were retested on day 21 on both Abbott RT and Aptima Assays. DBS was packaged with desiccants and stored at room temperature for the 21 days duration of the DBS stability study.

### Study population

The study enrolled a cross-section of HIV positive adults receiving care and treatment in health facilities in Nairobi and Busia and who gave written informed consent. The study participants were mainly patients whose VL was being monitored because they were on antiretroviral therapy and a very few who were yet to initiate treatment. Plasma collected from both sites were tested in both assays. Venous DBS tested in this study were prepared from blood collected from a subset of study participants in Nairobi. Due to this there is a difference in number of plasma and DBS samples that were tested. A small subset of DBS samples did not have the paired plasma results due to a failed run on m2000.

### Sample collection and preparation

Whole blood was collected from 2227 participants by phlebotomy, shipped to KEMRI HIV laboratories within 6 hours of collection, and processed within 12 hours of receipt. A total of 1312 samples provided only plasma. Paired plasma and DBS samples were prepared from a further 915 venous samples. To prepare plasma, the whole blood samples were centrifuged at 1,100g for 10 minutes within 24 hours of collection, and the plasma was stored at -80˚C. To prepare DBS, 70μl of venous blood was spotted in each of ten spots (2 DBS cards) per patient. The DBS samples were allowed to dry overnight. The first DBS card for each patient was used for day 0 testing, while the DBS cards for the 21-day time point were packaged with desiccants and stored at room temperature. All samples were de-identified prior to Aptima testing.

### Laboratory methods

All plasma and DBS samples were first tested using the Abbott platform [27]. Patient results were generated from the plasma samples tested on the Abbott platform. All DBS testing in

Abbott was performed with 2 spots. Briefly, each of two spots was gently pushed out of the DBS card into a centrifuge tube, which was then filled with 1.3 mL of *m*DBS buffer. Centrifuge tubes were incubated at room temperature for 30 min. Thereafter, 1 mL of eluent was pipetted out of each centrifuge tube into a secondary tube for testing according to the manufacturer's instructions [27]. To determine the performance of Aptima assay, both plasma and venous DBS were tested on Panther system [8, 9]. All DBS testing in Aptima was performed with one spot. In each case, 1 mL of DBS extraction buffer was added into a specimen aliquot tube after which one spot was carefully introduced into the tube. Specimen aliquot tubes were rocked at room temperature for 30 minutes, centrifuged for 2 min at 3000 rpm, arranged on a rack and introduced into the panther machine for processing. For plasma samples, 0.75mL was aliquoted into secondary Aptima specimen aliquot tubes, loaded onto racks and introduced into the panther machine for processing [8, 9]. The system draws 0.5mL from each tube for the assay. Fifteen tubes were loaded onto each rack, for a maximum of 6 racks. The seventh rack was loaded with four samples only while the eighth rack was loaded with a Negative Control, a Low Positive Control, a High Positive Control and a Calibrator. The racks were transported into their appropriate lanes, the bay doors closed, and processing initiated. Initial results were available in 3.5 hours, with five results received every five minutes thereafter. Results were posted as either "Not Detected" or "Invalid", or as "copies/mL".

To assess the effect stability of HIV-1 RNA in DBS over time, results of DBS tested on day 0 were compared with DBS tested on day 21 in Aptima. For the day 0 condition, freshly prepared DBS was tested. DBS packaged with desiccants in a Ziploc bag and stored at room temperature was tested for the 21-day stability timepoint.

Plasma results for Abbott and Aptima were reported without conversion. For the Abbott two spot DBS protocol, a conversion factor was applied by the software to convert results to copies/ mL of HIV-1. For Aptima DBS, a conversion factor was applied by the research team to convert results to copies/ mL of HIV-1. The linear quantitative range for Aptima using DBS was 883–10,000,000 copies/mL, while for the Abbott RT assay for plasma sample for 1 mL and 0.6mL protocol is 40–10,000,000 copies/mL, while that for the 0.5 mL protocol is 75–10,000,000 copies/mL. For this study, testing was performed with 0.6 mL of plasma sample. The linear quantitative range of Abbott Real-Time assay using the two spot DBS protocol is 550–10,000,000 copies/mL.

### Data analysis

Only valid results were used for analyses. VL data were transformed into $\log_{10}$ copies/mL. Statistical analysis was performed using Stata/MP version for Mac.

In the correlation analysis, only the VL data from patients who had quantitative values in both Aptima Assay and Abbott RT were included for analysis. The correlation was determined by simple linear regression with generation of Pearson's correlation coefficient (*r*) as well as Bland–Altman analysis [28] to calculate the bias and the limit of agreement between assay results. The agreement between the assays for the plasma and DBS sample types were determined at the clinical decision point of 1000 copies/mL using a contingency table.

## Results

### Test outcomes

A total of 2227 venous samples were successfully collected. For the plasma study, 1312 plasma samples were tested on both the Abbott and Aptima Assays. On the Abbott RT Assay, 713 samples returned "not detected" results. One sample failed while 598 samples had detectable

VLs; of those, 154 had <75 copies/mL while one (1) had <40 copies/mL. On the Aptima assay, 965 samples had quantifiable results while 347 returned "not detected" results. Of the quantifiable results, 451 had <30 copies/mL, while one sample had >10,000,000 copies/mL.

For the DBS study, 915 samples were tested on both assays. In all, 693 DBS samples had detectable VLs on the Abbott RT assay: of those, 6 had <550 copies/mL. A further 222 samples returned "not detected" results. For the Aptima Assay, 866 samples had quantifiable VLs on DBS. Of those, 213 had 870copies/mL while 3 had 10,000,000 copies/mL. Only 49 samples had "not detected" results.

## Agreement between Aptima and Abbott RT VL assay for various sample types at the clinical cut-off of 3 log copies/mL (1,000 copies/mL)

Plasma HIV-1 VL was conducted successfully for 1311 of the 1312 patients in both Aptima and Abbott RT assays and the results used for comparison. The positive agreement was 98.42% (95% CI 96.35%-99.32%), the negative agreement 97.19% (95% CI 95.97–98.05) and the overall agreement 97.48%. Fresh DBS from 915 patients were also tested and the results compared between the two assays. The positive agreement was 95.35% (95% CI 93.47%-96.70%) while the negative agreement was 92.77% (95% CI 88.86%-95.38%). The overall agreement was 94.64%. Fresh DBS was compared with three weeks old DBS for 910 patients to evaluate the stability of HIV-1 RNA in DBS; the positive agreement was 95.05% (95% CI 93.1%-96.5%) while the negative agreement was 97.34% (95%CI 94.6%-98.7%). The overall agreement was therefore 95.71% (Table 1).

The discordant rates between the two assays were 2.52% (33/1312) and 5.36% (49/915 samples) for plasma and DBS respectively. Of the 33 discordants for plasma, 28 had <1000 copies/mL for Abbott and >1000 copies/mL for Aptima while 5 samples had VL >1000 copies/mL in Abbott and <1000 copies/mL in Aptima.

Of the 49 patients with discordant DBS results, 31 were above MDP for Abbott RT and below MDP for Aptima with the reverse being true for the remaining 18 patients. A total of

**Table 1. Agreement between Aptima and Abbott RT VL assay at a clinical cut-off of 3 log copies/mL (1,000 copies/mL) of HIV for the various sample types.**

| | | | | | Result (95%CI) |
|---|---|---|---|---|---|
| | **Abbott RT Plasma** | | | | |
| **Aptima Plasma** | **<3.0** | **>3.0** | **Total** | | |
| **<3.0** | 968 (**97.19%**) | 5 (1.58%) | 973 (74.16%) | Negative Agreement | **97.19%** (95.97%-98.05%) |
| **>3.0** | 28 (2.81%) | 311 (**98.42%**) | 339 (25.51%) | Positive Agreement | **98.42%** (96.35%-99.32%) |
| **Total** | 996 (75.91%) | 316 (24.09%) | 1,312 (100.00%) | **Total Agreement** | 97.48% (96.49%-98.20%) |
| | **Abbott RT DBS** | | | | |
| **Aptima DBS** | **<3.0** | **>3.0** | **Total** | | |
| **<3.0** | 231 (**92.77%**) | 31 (4.58%) | 262 (28.63%) | Negative Agreement | **92.77%** (88.86%-95.38%) |
| **>3.0** | 18 (7.23%) | 635 (**95.35%**) | 653 (71.37%) | Positive Agreement | **95.35%** (93.47%-96.70%) |
| **Total** | 249(27.21%) | 666(72.79%) | 915 (100.00%) | **Total Agreement** | **94.64%** (92.99%-95.93%) |
| | **Aptima DBS day 0** | | | | |
| **Aptima DBS day 21** | **<3.0** | **>3.0** | **Total** | | |
| **<3.0** | 256 (**97.34%**) | 32 (4.95%) | 288 (31.65%) | Negative Agreement | **97.34%** (94.61%-98.70%) |
| **>3.0** | 7 (2.66%) | 615 (**95.05%**) | 622 (68.35%) | Positive Agreement | **95.05%** (93.10%-96.48%) |
| **Total** | 263 (28.90%) | 647 (71.10%) | 910 (100.00%) | **Total Agreement** | 95.71% (94.19%-96.85%) |

17/31 (54.84%) patients with Aptima DBS <3.0 and Abbott DBS >3.0 log copies/mL had a plasma VL <3.0 log copies/mL (1000 copies/mL) in both Aptima and Abbott assays. For 15 of these 17 patients, DBS VL was <3.0 log copies/mL on testing on day 21 in Aptima as well. For the remaining 14/31 patients, the plasma VL was between >3.0 log copies/mL in Aptima, with 12 of them being >3.0 in Abbott assay. Of the 18 patients with discordant results with Aptima DBS VL >3.0 and Abbott DBS VL <3.0 log copies/mL on day 0, 8 (44.4%) had Aptima DBS VL >3.0 log copies/mL on day 21.The remaining 10 (55.56%) had Aptima DBS VL <3.0 log copies/mL on day 21. The Aptima plasma VL was >3.0 log copies/mL for the 8 samples and <3.0 log for all the 10 samples.

## Regression analysis of the Aptima assay for VL testing

Four hundred and thirty-six (436) plasma samples had quantifiable results on both platforms and were used for correlation analysis. A significant correlation was found with a slope of 1.035 and *r* of 0.9589 (Fig 1).

Bland-Altman analysis yielded a mean bias of 0.33 log copies/mL with Aptima VL results being higher than Abbott RT. The upper and lower limits of agreement were 1.018 to -0.359; 7.11% of the comparisons lay outside the limits of agreement (Fig 2).

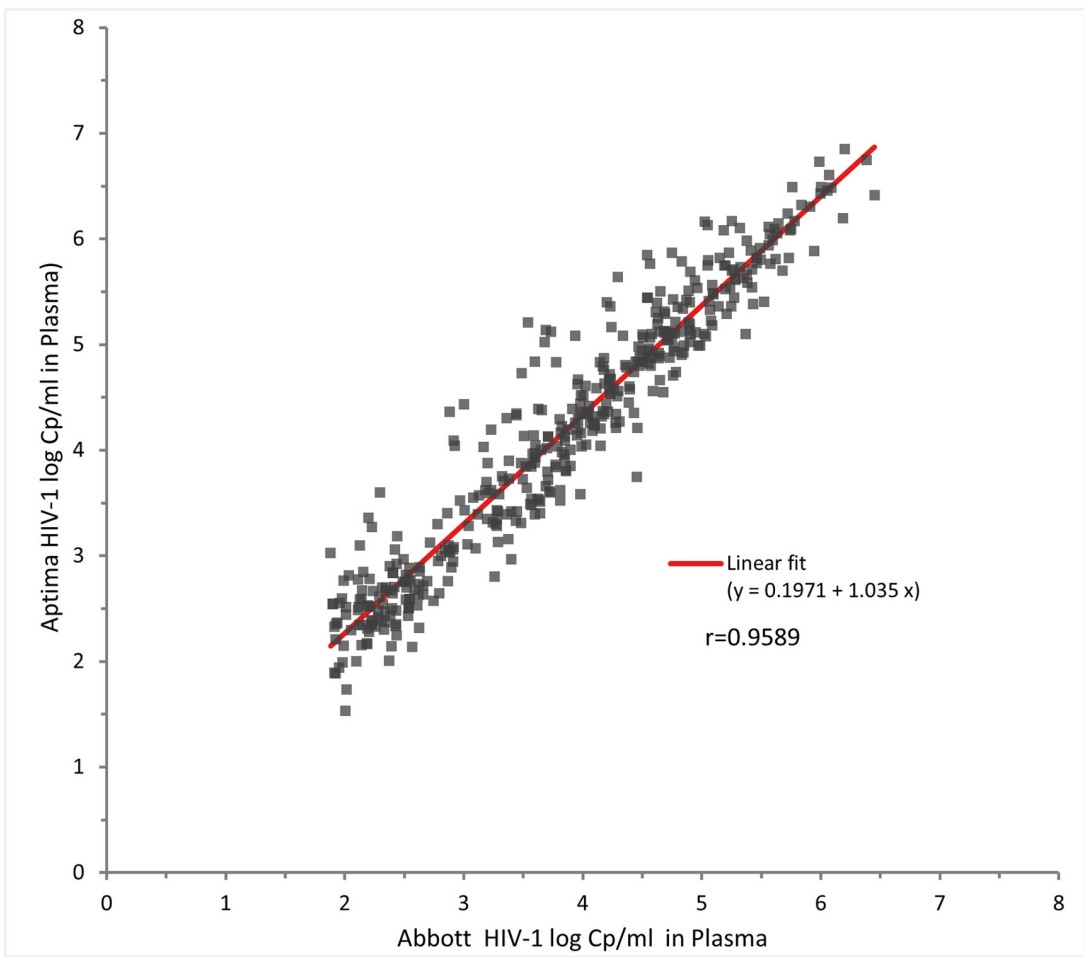

**Fig 1. HIV VL comparison for 436 plasma samples tested in Aptima and Abbott RT assays.**

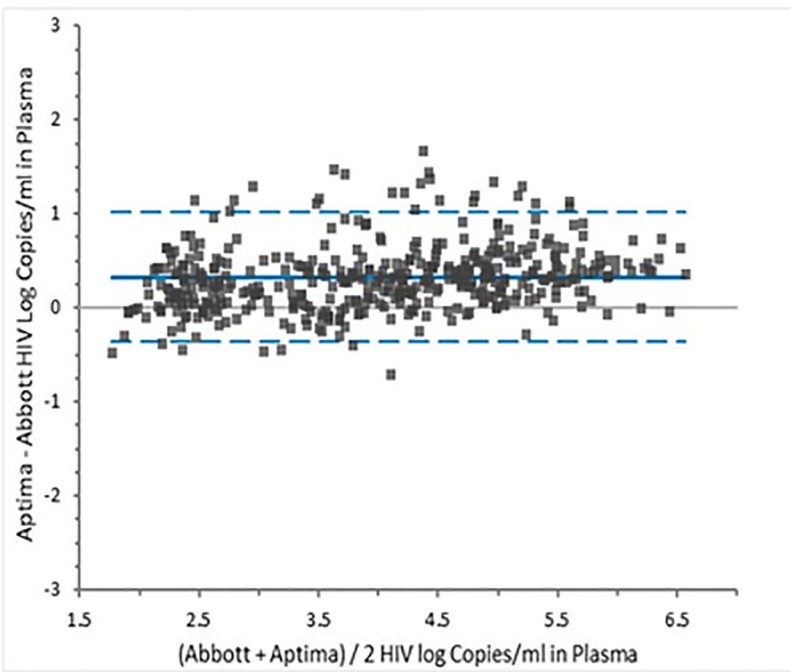

**Fig 2. Bland Altman analysis of HIV VL results for 436 plasma samples tested in Aptima and Abbott RT assays.**

## Comparison of fresh DBS tested on Aptima DBS and Abbott RT assays

Six hundred and thirty-six (636) fresh DBS samples had quantifiable results in both assays and were used for correlation analysis. On correlation, the slope was 0.9175 while *r* was 0.8692 (Fig 3).

Bland-Altman analysis yielded a mean bias of 0.35 log copies/mL (95% CI 1.04 and -0.34) across the assay range; 4.72% of the comparisons lay outside the limits of agreement (Fig 4).

## Fresh DBS compared with 421-day old DBS on the Aptima Assay

Six hundred and three (603) samples had quantifiable DBS results on day 0 and day 21. On correlation, the slope was 0.9328 while *r* was 0.9339 (Fig 5).

Bland-Altman analysis yielded a mean bias of -0.154 log copies/mL. The upper and lower limits of agreement were -0.644 to 0.336; 5.97% of the comparisons lay outside the limits of agreement (Fig 6).

## Discussion

To meet UNAIDS 90-90-90 treatment goals countries with high HIV burden need to perform millions of tests each year for HIV diagnosis and monitoring [1, 2]. The 10 centralized HIV testing labs in Kenya are often unable to meet this demand resulting in sample backlogs and long turnaround times to results because the equipment's they use for testing are not high throughput. In this study we evaluated the Aptima Assay for use in Kenya not only because it is an automated high throughput assay but also because it recently received CE IVD and WHO approval for HIV VL monitoring with both plasma and DBS specimens.

We observed a total agreement of 97.48% between Aptima and Abbott RT at the medical decision point of 1000 copies/mL on testing plasma specimens. In this comparison, both the

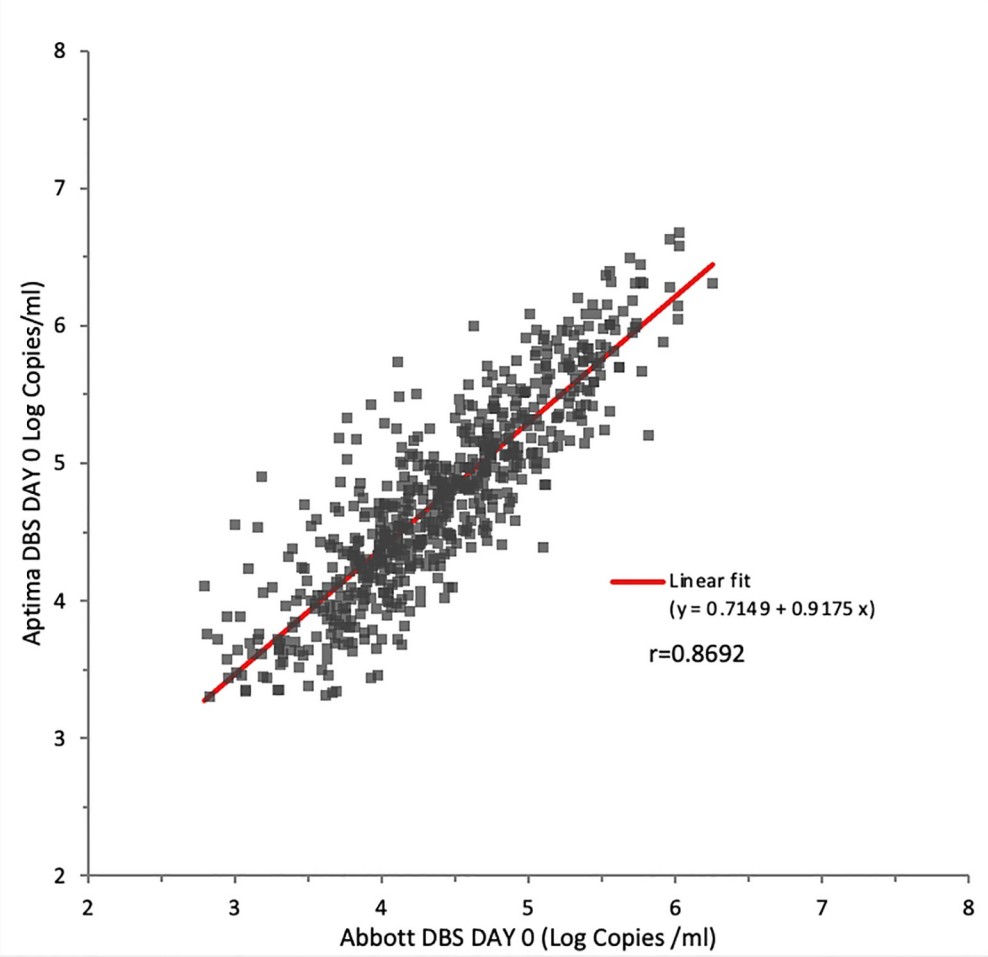

**Fig 3. HIV VL comparison in fresh DBS tested in Aptima and Abbott RT assays.**

positive and negative agreement were greater than 95%. Overall, the assay met the requirement of concordance at the clinical cut-off of 1000 copies/mL. However, there were 33 specimens out of 1312 (2.5%) with discordant results at the MDP of 1000 copies/mL. A majority of these (28) had HIV VL >1000 copies/mL in Aptima but < 1000 copies/mL in Abbott RT. Only 5 specimens had VL >1000 copies/mL in Abbott RT and <1000 copies/mL in Aptima of which 3 had results that were within 0.5 log of each other in the two assays. Discordant resolution was not performed for these specimens due to sample volume limitations.

Linear regression showed excellent agreement in plasma VL across the assay range with an *r* of 0.9589, slope of 1.035 and intercept of 0.1971. The slope is similar to that reported between Aptima and Abbott RT by other investigators in Europe [11, 20] and the United States [17, 18]. Bland Altman analysis showed that the VL in Aptima was 0.33 logs higher than Abbott RT which is in line with the 0.11 to 0.30 logs difference reported between the two assays by other investigators [11, 14, 17, 18]. Most Kenyan patients are infected with HIV subtype A or C and other investigators have reported a slightly higher bias between the two assays for these sub-types [11, 17, 18].

Currently, approximately 20% of the VL monitoring in Kenya is conducted with DBS speci-mens most of which are tested using the Abbott RT assay. The overall agreement between

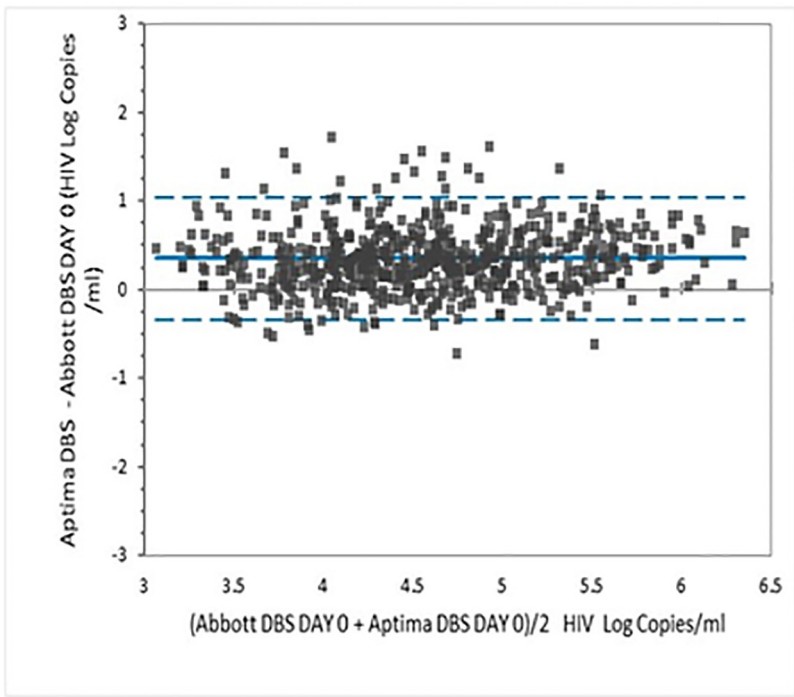

**Fig 4. Bland Altman analysis of HIV VL results in fresh DBS tested in Aptima and Abbott RT assays.**

Aptima and Abbott RT assays using DBS was 94.64% with positive and negative agreements of 95.35% and 92.77% respectively. Linear regression gave a slope of 0.9175 and an *r* of 0.8692. The 0.3 log bias seen between Aptima and Abbott for plasma specimens was also seen with DBS.

Although discordant resolution was not performed by testing the DBS is an independent assay, it was possible to assess the accuracy of DBS quantification in both Aptima and Abbott RT in much more detail by comparing it to the VL result of plasma from the same patient, tested in both assays. Of the 49 discordant DBS results between Aptima and Abbott RT around the MDP of 1000 copies/mL, 31 had results below MDP in Aptima and above MDP for Abbott RT while the reverse was true for the remaining 18 samples. Among the 31 patients with Abbott DBS VL >3 log copies/mL and Aptima DBS VL <3 log copies/mL, 17 of the patients had VL <3 log copies/mL in both Aptima plasma and Abbott plasma suggesting that these were upward misclassifications by Abbott DBS. The remaining 14 of the 31 patients with Aptima DBS VL <3 log copies/mL and Abbott DBS VL>3 log copies/mL had plasma VL >3.00 log copies/mL in both Aptima plasma and Abbott plasma suggesting that these were upward misclassifications by Abbott DBS.

Among the 18 discordant samples with Abbott DBS VL <3 log copies/mL, 8 of them had Aptima plasma and DBS results >3 log copies/mL. Five of the 8 also had Abbott plasma VL >3 log copies/mL. This suggests that these samples are likely to be downward misclassifications (under quantification) by Abbott DBS. Notably, among the 31 patients with Abbott DBS VL >3 log copies/mL and Aptima DBS VL < 3 log copies/mL, 17 of the patients also had VL <3 log copies/mL in both Aptima plasma and Abbott plasma suggesting that these were upward misclassifications by Abbott DBS. From these two situations, it can be concluded that significant variability may occur while testing DBS with concentrations close to the MDP of 1000 copies/mL in this study, regardless of the assay. This calls for further research, even

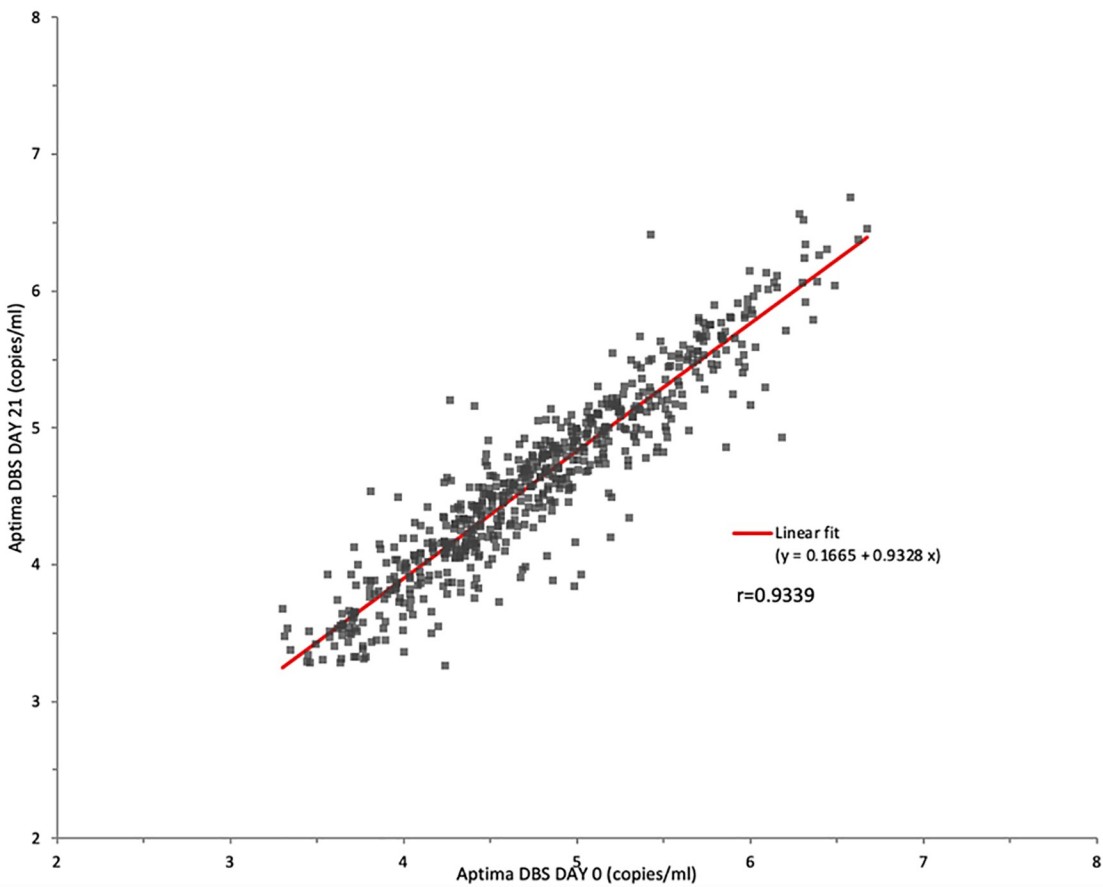

**Fig 5. HIV VL in DBS tested on day 0 and day 21 in the Aptima assay.**

though testing using a single replicate for DBS appears to be a contributing factor. Nonetheless, the overall misclassification rate for Aptima DBS (24/49) was comparable to that seen with Abbott DBS (25/49) at the MDP of 1000 copies/mL.

Although only venous DBS were tested in this study, we previously compared the recovery of HIV-1 from venous DBS, fingerstick DBS, and plasma with the Aptima test [23] showing good agreement between plasma, venous DBS, and fingerstick DBS. One limitation of the current study is that no replicate testing or retesting of invalid specimens was performed due to volume limitation for these clinical samples.

The agreement on paired DBS tested on day 0 versus after storage at room temperature for 21 days was 96.48%. There was excellent correlation ($r = 0.9339$) and minimal mean bias (-0.154 logs). This indicates that there was no significant degradation of HIV-1 RNA in DBS even after three weeks of storage at room temperature with desiccants. This is relevant because it often takes up to 3 weeks for DBS specimens to be delivered to the laboratory for testing. This duration of stability is in line with Hologic's claims that HIV-1 RNA is stable in DBS stored at 30 and 40˚C with relative humidity of up to 85% for at least 2 weeks [9]. Monleau et al has reported that DBS is stable for 3 months at 20˚C but it is not clear whether the relative humidity for the conditions of storage was reflective of the high relative humidity seen in Nairobi, Kenya [29].

The data that we present is relevant because there is no published data currently available comparing the VL in Aptima with Abbott RT assay at the medical decision point of 1000

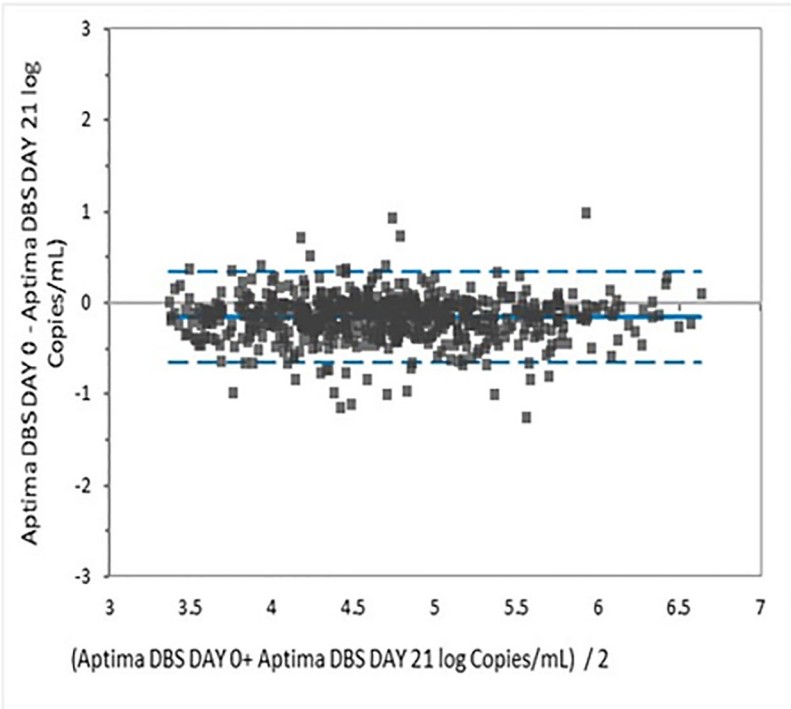

**Fig 6. Bland Altman analysis of HIV VL in DBS tested on day 0 and day 21 in the Aptima assay.**

copies/mL for both plasma and DBS specimens. The data in this publication generated show that there is excellent agreement between the two assays at 1000 copies/mL with plasma from over 1300 Kenyan patients. The same applies for the comparison of DBS specimens tested in the two assays. This close agreement with the Abbott RT assay implies that the Aptima Assay can be used interchangeably with Abbott RT assay for HIV VL determination in plasma and DBS. The excellent performance of Aptima for both plasma and DBS sample types meet the WHO recommendations for performance for VL monitoring [30].

Apart from the good performance, we observed some operational advantages in using the Aptima assay in the laboratory workflow during the study. Aptima is run on the Panther instrument platform that is fully automated and allows random and continuous loading of test samples. It allows loading of up to 515 tests in 12 hours, with the first results being returned in about 3.5 hours. In contrast, the Abbott m2000 platform allows 186 results in 10 hours, with the first results being returned in about 5 hours. This enables high flexibility to adapt to low or high-throughput testing. The one metre square footprint of Panther also makes it a good fit for laboratories with space constraints.

## Conclusions

In conclusion the performance of Aptima HIV-1 Quant Dx Assay was equivalent to that of Abbott RT for quantification of both plasma and DBS specimens from patients in Kenya. The high throughput, random access and complete automation provided by the Panther system enabled us to scale up testing up to 1000 specimens per day. This makes the Aptima Assay on Panther a good solution for clinical laboratories that need to close the gap between current testing capacity and UNAIDs goals for HIV-1 testing. It can help meet the increased demand for testing in any laboratory that performs VL testing.

## Acknowledgments

We thank the research assistants from the KEMRI HIV laboratory for their support.

## Author Contributions

**Conceptualization:** Matilu Mwau, Sven Schaffer.

**Data curation:** Matilu Mwau.

**Formal analysis:** Matilu Mwau, Sven Schaffer.

**Funding acquisition:** Matilu Mwau, Sven Schaffer.

**Investigation:** Matilu Mwau, Humphrey Kimani, Francis Ogolla, Elizabeth Ajema, Scriven Adoyo, Ednah Nyairo, Norah Saleri.

**Methodology:** Matilu Mwau, Sven Schaffer, Francis Ogolla, Elizabeth Ajema, Scriven Adoyo, Ednah Nyairo, Norah Saleri.

**Project administration:** Matilu Mwau.

**Resources:** Matilu Mwau, Sven Schaffer.

**Supervision:** Matilu Mwau.

**Validation:** Matilu Mwau.

**Visualization:** Matilu Mwau, Humphrey Kimani, Sangeetha Vijaysri Nair.

**Writing – original draft:** Matilu Mwau, Sangeetha Vijaysri Nair.

**Writing – review & editing:** Matilu Mwau, Humphrey Kimani, Purity Kasiano, Sangeetha Vijaysri Nair.

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
