## [Decision Letter · Decision Letter 0]

22 Oct 2021

PONE-D-21-27672Comparison of the performance of Aptima HIV-1 Quant Dx Assay with Abbott RealTime HIV Assay for viral load monitoring using plasma and Dried Blood Spots collected in KenyaPLOS ONE

Dear Dr. Mwau,

Thank you for submitting your manuscript to PLOS ONE. After careful consideration, we feel that it has merit but does not fully meet PLOS ONE’s publication criteria as it currently stands. Therefore, we invite you to submit a revised version of the manuscript that addresses the points raised during the review process.

Two reviewers and I have provided comments that must be addressed for further consideration of the manuscript, especially in addressing the lack of replicate testing and problems with the statistical analysis.

We look forward to receiving your revised manuscript.

Kind regards,

Julie AE Nelson, PhD

Academic Editor

PLOS ONE

Additional Editor Comments (if provided):

The major issues to address are the inconsistent numbers throughout, the lack of replicate testing, and the higher rate of quantitative values from DBS than from plasma. These and other issues are detailed below:

Line 17: The background section is too long—drop one or two of the sentences.

Line 32: Add that venipuncture was the source of blood for the DBS in this study.

Line 33: Use the correct name of the Abbott assay: Abbott RealTime HIV-1 Assay

Line 37: Drop the sentence about the Stata software, since that is not needed in the abstract.

Make sure everything is the same between the text and the abstract and the figures (436 vs 439; r = 0.96 vs R = 0.92—need to check both the number and the capitalization for that one; r = 0.87 vs R = 0.8692—need to check number of significant digits and capitalization; mean bias for DBS storage = -0.154 vs -0.15). Be consistent with your significant digits and with small case r throughout.

Line 46: Drop the sentence about the discordance percentages since they are just obtained by subtracting the agreement percentages from 1 and provide nothing new.

Line 48: Remove this last sentence of the results about the Panther platform throughput—not needed in the abstract.

Line 52: this section needs more about the interpretation of the data in the results section and could mention that the assay worked well on samples in Kenya specifically.

Line 74: small case m2000

Line 75: there is a (ref) that needs the citation added.

Line 79: add the full name of the Aptima assay here: Hologic Aptima HIV-1 Quant Dx Assay.

Line 110: Fix the name of the Abbott assay: Abbott RealTime HIV-1 Assay.

Line 136: clarify what is meant by testing DBS on day 0. Is that the day after spotting with no time in the presence of desiccants?

Line 144: the lower limit for the Abbott assay is 40 for both the 1mL and 0.6mL protocols, and 75 for the 0.5mL protocol, but all the linear ranges can be removed from the methods except possibly for the DBS protocols for each assay. Since all the analysis is with the 1000cp/mL cut-off, the lower limits for plasma are not needed. However, it would be useful to specify which volume protocol was used for the Abbott plasma testing since that is not stated.

Line 167: The use of positive predictive values and negative predictive values in this context is incorrect since there does not appear to be a “gold standard” to show which samples should be positive and which should be negative, suggesting that one of the assays was considered the gold standard for the calculations. All these calculations should be removed from the manuscript.

Line 169: There is a difference in number of plasma and DBS samples included in the data—add an explanation of the difference, whether this was because DBS hadn’t been made for some, the DBS data were invalid, or what other reason.

Line 176: remove the word “therefore”.

Line 178: remove the sentence about discordance rate

Line 184: discordance rates not shown in Table 1 but can be obtained with the positive agreement values in Table 1, so re-write this statement. This paragraph (lines 184-191) is difficult to understand in terms of useful results. It is useful to the reader to know how many of the discordant results were due to higher detection in one assay vs the other, but the value of the rest of the “results” is less clear. What is the goal of reporting that 17 of the discordant samples (Abbott higher than Aptima in DBS) had corresponding plasma VL that were higher than 3631cp/mL? Is that VL cutoff a magic number for these assays? There is no mention of these “results” in the discussion, providing no context for why this “result” should be noted.

Table 1: the two sides of the table are not related because different numbers of samples are used for each side. Therefore, remove the agreement data from the table and make a new table of these values, adding the number of samples included in each calculation. Also add the DBS day 21 calculations that are not shown in the current table. As mentioned above, remove the PPV and NPV data.

Line 277: add the daily throughput of the Abbott m2000 as a comparator here to strengthen your argument that the Panther has higher throughput. Alternatively, add the throughput (tests per day) in line 75 where you describe it as medium throughput.

Line 218: the discussion needs to start with discussing the results that have just been presented in the previous section, so move the first paragraph down. Reorganize the discussion to move the summary of the results all to the top paragraph(s), then start to discuss them with other published studies in subsequent paragraphs.

Line 232: remove the discordance rate here since it is essentially a retelling of the initial result of agreement.

Line 234: change “several” to the actual number of these specimens. This is also a good place to add discussion of the higher number of DBS with quantifiable VL than matched plasma with quantifiable VL, which is a critical question in this study.

No replicate testing is mentioned, not even for invalid results. When running assays on samples that are near the lower limit of detection/quantitation, replicate testing can be critical. Therefore, this must be discussed as a limitation of the study in the discussion.

Check all the references that they are cited in the manuscript. Your previous publication in PLoS One from April is highly relevant, especially when discussing the limitation that you did not compare fingerstick DBS since you previously looked at that in the April publication.

Add the versions and dates used the package inserts in references 8 and 26.

Figures 1-3: All the (a) figure legends must be changed to be more like the (b) legends where the “Figure 1a” is not part of the title and contain sufficient information to stand alone. For example: Figure 1a: HIV-1 viral load comparison for 439 plasma samples tested on both the Aptima HIV-1 Quantitative RNA assay and the Abbott HIV-1 RealTime assay.

Journal Requirements:

a) Did participants provide their written or verbal informed consent to participate in this study?

3. Thank you for stating the following in the Competing Interests/Financial Disclosure * (delete as necessary) section:

“Hologic, Inc provided the funds and materials for the study.”

We note that you received funding from a commercial source: “Hologic, Inc.”

“Sangeetha Nair and Sven Schaffer were employees of Hologic, Inc at the time of the study. The rest of the authors, based at Kenya Medical Research Institute, declare no conflict of interest.”

Reviewers' comments:

Reviewer's Responses to Questions

**Comments to the Author**

1. Is the manuscript technically sound, and do the data support the conclusions?

Reviewer #1: Yes

Reviewer #2: No

2. Has the statistical analysis been performed appropriately and rigorously? 

Reviewer #1: Yes

Reviewer #2: Yes

3. Have the authors made all data underlying the findings in their manuscript fully available?

Reviewer #1: Yes

Reviewer #2: No

4. Is the manuscript presented in an intelligible fashion and written in standard English?

Reviewer #1: Yes

Reviewer #2: Yes

5. Review Comments to the Author

Reviewer #1: This is an easy to read manuscript providing important data relevant to increasing access to HIVVL testing in Africa. have the following comments.

line 55 - Update the HIV statistics to the most recent.

line 95 - I suggest "preferred reference in Kenya' rather than gold standard.

line 107 - Study Design -Was there sufficient plasma to test the same samples on both assays i.e. same plasma samples tested on Aptima and Abbott?

line 118 - Study Population- inclusion criteria needs to be more detailed, e.g., on ARVs or not, consecutive patients, routine clinic visits

line 127 - Was plasma separated on the same day as DBS preparation?

line 132 - How soon after collection were the Abbott tests performed?

line 134- What is the time difference between Aptima and Abbott tests?

Some detail is required on the Aptima procedure as this is the "study" assay.

line 142 - Input volume for the Aptima ?

line 144 - Which protocol was used for the Abbott?

line 167 - Results - was prevalence taken into account for the PPV and NPV?

line 184 - It maybe easier to describe discordance as upward and downward misclassification and the range of the VL obtained for the misclassified samples according to sample type.

lines 281 - 226 -Discussion - This part is more suited to the introduction.

line 234 - is there an explanation for this finding?

line 242 - is there a reason for this observation?

line 250 - is there a reason for this limitation?

line 277 - is this per 8/12/24 hrs? This will give a better idea of throughput.

Reviewer #2: Mwau et al. have investigated the performance of the aptima HIV1 assay compared to the Abbott real-time assay on plasma and DBS samples, at the 1000cp/mL decision point, as well as data on HIV RNA stability at room temperature. 1312 paired plasma and venous blood DBS samples were used. Total agreement between Abbott and Aptima assays at the 3 log threshold was 97.48% for plasma and 94.64% for DBS. Data on quantification agreement between assays, and over time for DBS are provided.

Major points

It is not clear at abstract’s reading that authors were working on DBS prepared from venous blood and not from finger-stick. The number of quantifiable samples is different in abstract 436 and results 439. The direction of the quantification bias (higher in aptima or abbott) is not clear. % of agreements are given but which % is needed to say that an assay is accurate?

In the background, authors acknowledge that several papers have been published on the aptima assay performance compared to other platforms on plasma samples, but not assessing agreement at the medical decision threshold. However, authors have published such a study in Pone in 2021, with the same ethics number, though with fewer samples. In that study, authors have already evaluated FS and VB DBS samples. This work is not discussed, nor referenced.

In the results total, positive and negative agreements between assays at the 1000 cp MDP are given, but there is no data on assay variability at this level of VL. This repeatability and reproducibility assessment is mandatory for the interpretation of assays’ discordances.

Evaluations of the aptima assay from DBS samples have been published, but authors did not discuss these previously published results, nor the limitations of HIV RNA testing from DBS at low viral loads where HIV DNA, of cell-associated DNA may lead to over quantification of VL.

439 quantifiable plasma VL but 636 quantifiable fresh VB DBS: this discrepancy is not discussed

Minor points

• The daily throughput of abbott and hologic platforms is not specified.

• Works of Yek, Carrera, and Sahoo appear in the reference list but are not cited in the text. These authors have published on DBS and on the aptima assay

• In the methods, a flow chat of tested samples is lacking

6. PLOS authors have the option to publish the peer review history of their article (what does this mean?). If published, this will include your full peer review and any attached files.

Reviewer #1: No

Reviewer #2: No

---

## [Author Response · Author response to Decision Letter 0]

10 Dec 2021

Additional Editor Comments (if provided):

The major issues to address are the inconsistent numbers throughout, the lack of replicate testing, and the higher rate of quantitative values from DBS than from plasma. These and other issues are detailed below: 

a. Corrected the inconsistent in N from 439 to 436 for quantified plasma samples throughout the document

b. Clinical studies like the one included in this paper do not have enough specimen volume to permit replicate testing in 2 different assays. Replicate testing is more commonly performed for panels than for clinical samples

c. The number of quantifiable results in plasma and DBS sample sets are different because not all plasma and DBS were prepared from the same patients. Blood samples from only a subset of patients were used to prepare plasma and DBS for testing in the two assays. The study also included some DBS samples that did not have corresponding plasma samples and vice versa. 

Line 17: The background section is too long—drop one or two of the sentences.

Deleted a couple of sentences from the background section

Line 32: Add that venipuncture was the source of blood for the DBS in this study.

Added the information that DBS was prepared using venous blood

Line 33: Use the correct name of the Abbott assay: Abbott RealTime HIV-1 Assay

Corrected the name

Line 37: Drop the sentence about the Stata software, since that is not needed in the abstract.

Removed the statement

Make sure everything is the same between the text and the abstract and the figures (436 vs 439; r = 0.96 vs R = 0.92—need to check both the number and the capitalization for that one; r = 0.87 vs R = 0.8692—need to check number of significant digits and capitalization; mean bias for DBS storage = -0.154 vs -0.15). Be consistent with your significant digits and with small case r throughout.

Changed correlation coefficient from “R” to “r” in the document

Line 46: Drop the sentence about the discordance percentages since they are just obtained by subtracting the agreement percentages from 1 and provide nothing new.

Removed the statement on discordant rate

Line 48: Remove this last sentence of the results about the Panther platform throughput—not needed in the abstract.

Removed the statement

Line 52: this section needs more about the interpretation of the data in the results section and could mention that the assay worked well on samples in Kenya specifically.

Added a few words to make the conclusion specific to samples in Kenya

Line 74: small case m2000

Changed case for m in m2000

Line 75: there is a (ref) that needs the citation added.

This was the scenario in 2018. It has been resolved by combining two sentences into one: “In Kenya, viral load tests are conducted using Abbott and Roche assays running on the m2000 or COBAS Ampliprep/COBAS Taqman platforms”. Line 70

Line 79: add the full name of the Aptima assay here: Hologic Aptima HIV-1 Quant Dx Assay.

Added the full name of the Aptima assay

Line 110: Fix the name of the Abbott assay: Abbott RealTime HIV-1 Assay.

Fixed the name of the Abbott RT assay

Line 136: clarify what is meant by testing DBS on day 0. Is that the day after spotting with no time in the presence of desiccants?

Yes. For the day 0 condition, freshly prepared DBS was used for testing while DBS was stored in the presence of dessicants in a Ziploc bag prior to testing on day 21

Line 144: the lower limit for the Abbott assay is 40 for both the 1mL and 0.6mL protocols, and 75 for the 0.5mL protocol, but all the linear ranges can be removed from the methods except possibly for the DBS protocols for each assay. Since all the analysis is with the 1000cp/mL cut-off, the lower limits for plasma are not needed. However, it would be useful to specify which volume protocol was used for the Abbott plasma testing since that is not stated.

This has been resolved. The section has been rewritten to read as follows: “The linear quantitative range for Aptima using DBS was 883-10,000,000 copies/mL, while for the Abbott RT assay for plasma sample for 1 mL and 0.6mL protocol is 40–10,000,000 copies/mL, while that for the 0.5 mL protocol is 75-10,000,000 copies/mL. For this study, testing was performed with 0.6 mL of plasma sample. The linear quantitative range of Abbott Real-Time assay using the two spot DBS protocol is 550-10,000,000 copies/mL.”

The Abbott protocol used is 0.6mL (line 166-167)

Line 167: The use of positive predictive values and negative predictive values in this context is incorrect since there does not appear to be a “gold standard” to show which samples should be positive and which should be negative, suggesting that one of the assays was considered the gold standard for the calculations. All these calculations should be removed from the manuscript.

Removed NPV and PPV from this paragraph and Table 1

Line 169: There is a difference in number of plasma and DBS samples included in the data—add an explanation of the difference, whether this was because DBS hadn’t been made for some, the DBS data were invalid, or what other reason.

DBS was not collected for all the patients from whom plasma was collected.

Line 176: remove the word “therefore”.

Removed “therefore”

Line 178: remove the sentence about discordance rate

Removed the sentence on discordant rate

Line 184: discordance rates not shown in Table 1 but can be obtained with the positive agreement values in Table 1, so re-write this statement. This paragraph (lines 184-191) is difficult to understand in terms of useful results. It is useful to the reader to know how many of the discordant results were due to higher detection in one assay vs the other, but the value of the rest of the “results” is less clear. What is the goal of reporting that 17 of the discordant samples (Abbott higher than Aptima in DBS) had corresponding plasma VL that were higher than 3631cp/mL? Is that VL cutoff a magic number for these assays? There is no mention of these “results” in the discussion, providing no context for why this “result” should be noted.

Removed the reference to Table 1 from the statement of discordant rate. 

The purpose of pointing out that 31 samples with discordant DBS results at MDP of 1000 c/mL had plasma results <3631 c/mL is to demonstrate that these samples may flip above and below a 1000 c/mL in different replicates because the HIV concentration is close to a 1000 c/mL. Also added a couple of sentences under discussion to explain the DBS discordants to link with the results in this section.

Table 1: the two sides of the table are not related because different numbers of samples are used for each side. Therefore, remove the agreement data from the table and make a new table of these values, adding the number of samples included in each calculation. Also add the DBS day 21 calculations that are not shown in the current table. As mentioned above, remove the PPV and NPV data.

The two sides of Table 1 are related and use the same number of samples. The right side of Table 1 gives the positive agreement, negative agreement and total agreement for the results included in the 2x2 table on the left side of Table 1. The sample numbers on both sides of the table have been highlighted to show this and the corresponding results on each side of Table 1 have been moved to the same rows

DBS day 21 are already part of Table 1

Removed PPV and NPV data

Line 277: add the daily throughput of the Abbott m2000 as a comparator here to strengthen your argument that the Panther has higher throughput. Alternatively, add the throughput (tests per day) in line 75 where you describe it as medium throughput.

The Abbott m2000 platform allows 186 results in 10 hours, with the first results being returned in about 5 hours. This is clarified in lines 356-358

Line 218: the discussion needs to start with discussing the results that have just been presented in the previous section, so move the first paragraph down. Reorganize the discussion to move the summary of the results all to the top paragraph(s), then start to discuss them with other published studies in subsequent paragraphs.

Shortened the general information in the 1st paragraph 

Line 232: remove the discordance rate here since it is essentially a retelling of the initial result of agreement. 

Removed the discordance rate 

Line 234: change “several” to the actual number of these specimens. This is also a good place to add discussion of the higher number of DBS with quantifiable VL than matched plasma with quantifiable VL, which is a critical question in this study.

28 of the 1312 specimens had VL >1000 c/mL in Aptima and <1000 c/mL in Abbott RT including 5 that were not detected by Abbott RT. Five plasma specimens had a VL >1000 c/mL in Abbott RT and <1000 c/mL in Aptima including one sample that was not detected.

No replicate testing is mentioned, not even for invalid results. When running assays on samples that are near the lower limit of detection/quantitation, replicate testing can be critical. Therefore, this must be discussed as a limitation of the study in the discussion.

Added a limitation for not performing replicate testing and retesting of invalid specimens 

Check all the references that they are cited in the manuscript. Your previous publication in PLoS One from April is highly relevant, especially when discussing the limitation that you did not compare fingerstick DBS since you previously looked at that in the April publication.

Added a reference to Aptima DBS manuscript published in PLOS One in April 2021 titled “Prospective evaluation of accuracy of HIV viral load monitoring using the Aptima HIV Quant Dx assay with fingerstick and venous dried blood spots prepared under field conditions in Kenya” in the reference section (reference 31) and referenced it in the text in the discussion

Add the versions and dates used the package inserts in references 8 and 26.

Added version numbers for references 8 and 26 

Figures 1-3: All the (a) figure legends must be changed to be more like the (b) legends where the “Figure 1a” is not part of the title and contain sufficient information to stand alone. For example: Figure 1a: HIV-1 viral load comparison for 439 plasma samples tested on both the Aptima HIV-1 Quantitative RNA assay and the Abbott HIV-1 RealTime assay.

Updated Figure legends for Figure 1-3 

Journal Requirements:

https://journals.plos.org/plosone/s/file?id=wjVg/PLOSOne_formatting_sample_main_body.pdf [journals.plos.org] and

https://journals.plos.org/plosone/s/file?id=ba62/PLOSOne_formatting_sample_title_authors_affiliations.pdf [journals.plos.org]

a) Did participants provide their written or verbal informed consent to participate in this study?

The study enrolled a cross-section of HIV positive adults receiving care in health facilities in Nairobi and Busia and who gave written informed consent. Please see Line 121 and 122.

We took written informed consent.

3. Thank you for stating the following in the Competing Interests/Financial Disclosure * (delete as necessary) section:

“Hologic, Inc provided the funds and materials for the study.”

We note that you received funding from a commercial source: “Hologic, Inc.”

Within this Competing Interests Statement, please confirm that this does not alter your adherence to all PLOS ONE policies on sharing data and materials by including the following statement: "This does not alter our adherence to PLOS ONE policies on sharing data and materials.” (as detailed online in our guide for authorshttp://journals.plos.org/plosone/s/competing-interests [journals.plos.org]). If there are restrictions on sharing of data and/or materials, please state these. Please note that we cannot proceed with consideration of your article until this information has been declared.

As stated above, Hologic, Inc. provided the funds and materials for the study. This does not alter our adherence to PLOS ONE policies on sharing data and materials. Sangeetha Nair and Sven Schaffer were employees of Hologic, Inc. at the time of the study. This does not alter our adherence to PLOS ONE policies on sharing data and materials. In addition, there are restrictions on the sharing of data and materials originating from the study. The rest of the authors are employees of the Kenya Medical Research Institute and declare no conflict of interest.

We have put this statement in the cover letter.

“Sangeetha Nair and Sven Schaffer were employees of Hologic, Inc at the time of the study. The rest of the authors, based at Kenya Medical Research Institute, declare no conflict of interest.”

Please confirm that this does not alter your adherence to all PLOS ONE policies on sharing data and materials, by including the following statement: "This does not alter our adherence to PLOS ONE policies on sharing data and materials.” (as detailed online in our guide for authors http://journals.plos.org/plosone/s/competing-interests [journals.plos.org]). If there are restrictions on sharing of data and/or materials, please state these. Please note that we cannot proceed with consideration of your article until this information has been declared.

As stated above, Hologic, Inc. provided the funds and materials for the study. This does not alter our adherence to PLOS ONE policies on sharing data and materials. Sangeetha Nair and Sven Schaffer were employees of Hologic, Inc. at the time of the study. This does not alter our adherence to PLOS ONE policies on sharing data and materials. In addition, there are restrictions on the sharing of data and materials originating from the study. The rest of the authors are employees of the Kenya Medical Research Institute and declare no conflict of interest.

We have put this statement in the cover letter.

5. Review Comments to the Author

Reviewer #1: This is an easy to read manuscript providing important data relevant to increasing access to HIVVL testing in Africa. I have the following comments.

line 55 - Update the HIV statistics to the most recent.

Updated the statistics and relevant reference

line 95 - I suggest "preferred reference in Kenya' rather than gold standard.

Updated this change

line 107 - Study Design -Was there sufficient plasma to test the same samples on both assays i.e. same plasma samples tested on Aptima and Abbott?

Yes.

line 118 - Study Population- inclusion criteria needs to be more detailed, e.g., on ARVs or not, consecutive patients, routine clinic visits. 

The section has been beefed up to read as follows:

The study enrolled a cross-section of HIV positive adults receiving care and treatment in health facilities in Nairobi and Busia and who gave written informed consent. The study participants were mainly patients whose viral load was being monitored because they were on antiretroviral therapy and a very few who were yet to initiate treatment. Plasma collected from both sites were tested in both assays for this study Venous DBS tested in this study were prepared from blood collected from a subset of study participants in Nairobi . Due to this there is a difference in number of plasma and DBS samples that were tested. A small subset of DBS samples did not have the paired plasma results due to a failed run on m2000.

line 127 - Was plasma separated on the same day as DBS preparation?

Yes. The venous blood was shipped to the lab where it was spotted onto DBS cards and then centrifuged to separate the plasma on the same day 

line 132 - How soon after collection were the Abbott tests performed?

Whole blood was collected by phlebotomy and shipped to KEMRI HIV laboratories within 6 hours of collection, and tested within 12 hours of receipt. Please see lines 132-133

line 134- What is the time difference between Aptima and Abbott tests?

The Aptima assay returned the first batch of results within 3.5 hours while the Abbott assay did so within 5 hours. Please see lines 356,357 and 358

Some detail is required on the Aptima procedure as this is the "study" assay.

The following text has been added from lines 146-154:

To determine the performance of Aptima assay, both plasma and venous DBS were tested on Panther system according to manufacturer’s instructions [8,9,]. Briefly, after initial preparation, 0.75mL of plasma or DBS samples were aliquoted into secondary Aptima specimen aliquot tubes. The system draws 0.5mL from each tube for the assay. Fifteen tubes were loaded onto each rack, for a maximum of 6 racks. The seventh rack was loaded with four samples only while the eighth rack was loaded with a Negative Control, a Low Positive Control, a High Positive Control and a Calibrator. The racks were transported into their appropriate lanes, the bay doors closed, and processing initiated. Initial results were available in 3.5 hours, with five results received every five minutes thereafter. Results were posted as either “Not Detected” or “Invalid”, or as “copies/mL”.

line 142 - Input volume for the Aptima ?

0.75 mL of sample is used per test as indicated in line 146-145. The system draws 0.5mL from each tube for the assay

line 144 - Which protocol was used for the Abbott?

We used the 0.6mL protocol for the Abbott test, as indicated in line 143. 

line 167 - Results - was prevalence taken into account for the PPV and NPV?

All mention of PPV and NPV has been removed from the manuscript

line 184 - It may be easier to describe discordance as upward and downward misclassification and the range of the VL obtained for the misclassified samples according to sample type.

Added this to the discussion for DBS results

lines 281 - 226 -Discussion - This part is more suited to the introduction.

Reduced the 1st paragraph of the discussion because a lot of this covered under introduction

line 234 - is there an explanation for this finding?

Yes. Additional sample was not available for discordant resolution

line 242 - is there a reason for this observation?

HIV-1 has 4 groups (M, N, O and P) with group M having at least 9 subtypes (A, B, C,D, F, G, H, J, K) and numerous recombinants with significant differences in their nucleic acid sequences. Due to this all viral load assays have slight differences in their ability to quantify each HIV subtype although they are optimized to quantify within 0.5 log of expected concentration. Similar differences have been demonstrated between Abbott RT and Roche Cobas Ampliprep Cobas Taqman assays for HIV (see reference 17 Manak et al)

line 250 - is there a reason for this limitation? 

The limitation regarding testing venous DBS instead of finger stick DBS has been removed from this publication. Instead, we have included a reference that shows equivalency of venous and fingerstick DBS results from the same patient on testing in Aptima

line 277 - is this per 8/12/24 hrs? This will give a better idea of throughput.

Up to 515 samples can be loaded within 12 hours for testing on Panther. Please see line 356.

Reviewer #2: Mwau et al. have investigated the performance of the aptima HIV1 assay compared to the Abbott real-time assay on plasma and DBS samples, at the 1000cp/mL decision point, as well as data on HIV RNA stability at room temperature. 1312 paired plasma and venous blood DBS samples were used. Total agreement between Abbott and Aptima assays at the 3 log threshold was 97.48% for plasma and 94.64% for DBS. Data on quantification agreement between assays, and over time for DBS are provided.

Major points

1. It is not clear at abstract’s reading that the authors were working on DBS prepared from venous blood and not from finger-stick. The number of quantifiable samples is different in abstract 436 and results 439. 

Added a statement in the methods that venous DBS was used for this study. Corrected the number of quantified samples to 436 

2. The direction of the quantification bias (higher in Aptima or Abbott) is not clear. % of agreements are given but which % is needed to say that an assay is accurate?

Added more discussion on the discordants. The discussion clarifies the number of discordant samples at the MDP of 1000 c/mL for each sample type

3. In the background, authors acknowledge that several papers have been published on the aptima assay performance compared to other platforms on plasma samples, but not assessing agreement at the medical decision threshold. However, authors have published such a study in Pone in 2021, with the same ethics number, though with fewer samples. In that study, authors have already evaluated FS and VB DBS samples. This work is not discussed, nor referenced.

Added a reference to the publication in PLOS One in 2021 comparing finger stick and venous DBS in Aptima

4. In the results total, positive and negative agreements between assays at the 1000 cp MDP are given, but there is no data on assay variability at this level of VL. This repeatability and reproducibility assessment is mandatory for the interpretation of assays’ discordances.

The repeatability and reproducibility results of the two assays for plasma and DBS is presented in the package inserts that are referenced

The reproducibility of Aptima and Abbott assays at 1000 c/mL is published in the package inserts of these assays which is referenced in this publication

5. Evaluations of the aptima assay from DBS samples have been published, but authors did not discuss these previously published results, nor the limitations of HIV RNA testing from DBS at low viral loads where HIV DNA, of cell-associated DNA may lead to over quantification of VL.

We had added a reference to our recent publication

Aptima HIV Quant Dx assay uses a technology that amplifies DNA at 10,000 to 100,000 fold lower efficiency than RNA. Due to this, no overquantification of VL has been observed in DBS samples. 

6. 439 quantifiable plasma VL but 636 quantifiable fresh VB DBS: this discrepancy is not discussed

The number of quantifiable results in plasma and DBS sample sets are different because plasma and DBS were not prepared from all the same patients. Blood samples from only a subset of patients were used to prepare plasma and DBS for testing in the two assays. The study also included some DBS samples that did not have corresponding plasma samples and vice versa. Statements to clarify this has been added to the methods section

Minor points

1. The daily throughput of Abbott and Hologic platforms is not specified.

The Aptima assay allows loading of up to 515 tests in 12 hours, with the first results being returned in about 3.5 hours. In contrast, the Abbott m2000 platform allows 186 results in 10 hours, with the first results being returned in about 5 hours. Please see lines 356-358.

2. Works of Yek, Carrera, and Sahoo appear in the reference list but are not cited in the text. These authors have published on DBS and on the aptima assay

References to these publications have been added in the “Background section”

3. In the methods, a flow chat of tested samples is lacking

A flow chart was not added because only the blood from a subset of the 1312 patients in the plasma method comparison study were used to prepare DBS. The DBS study also had some patients who did not have plasma results. Statements to clarify this has been added to the methods section

Reviewers' comments:

Reviewer's Responses to Questions

Comments to the Author

1. Is the manuscript technically sound, and do the data support the conclusions?

Reviewer #1: Yes

Reviewer #2: No

2. Has the statistical analysis been performed appropriately and rigorously?

Reviewer #1: Yes

Reviewer #2: Yes

3. Have the authors made all data underlying the findings in their manuscript fully available?

The PLOS Data policy [plosone.org] requires authors to make all data underlying the findings described in their manuscript fully available without restriction, with rare exception (please refer to the Data Availability Statement in the manuscript PDF file). The data should be provided as part of the manuscript or its supporting information, or deposited to a public repository. For example, in addition to summary statistics, the data points behind means, medians and variance measures should be available. If there are restrictions on publicly sharing data—e.g. participant privacy or use of data from a third party—those must be specified.

Reviewer #1: Yes

Reviewer #2: No

4. Is the manuscript presented in an intelligible fashion and written in standard English?

Reviewer #1: Yes

Reviewer #2: Yes

---

## [Decision Letter · Decision Letter 1]

16 Mar 2022

PONE-D-21-27672R1Comparison of the performance of Aptima HIV-1 Quant Dx Assay with Abbott RealTime HIV Assay for viral load monitoring using plasma and Dried Blood Spots collected in KenyaPLOS ONE

Dear Dr. Mwau,

Thank you for submitting your manuscript to PLOS ONE. After careful consideration, we feel that it has merit but does not fully meet PLOS ONE’s publication criteria as it currently stands. Therefore, we invite you to submit a revised version of the manuscript that addresses the points raised during the review process.

There are still many errors in the manuscript pointed out by Reviewer #1 that need to be addressed. Reviewer #2 was unavailable , so I have added comments from my review of the manuscript. Once the comments of these two reviews have been addressed, the manuscript will be much more clear to the reader. 

We look forward to receiving your revised manuscript.

Kind regards,

Julie AE Nelson, PhD

Academic Editor

PLOS ONE

Journal Requirements:

Additional Editor Comments:

Editor's review:

Line 30: remove the word consenting—this is not needed in the abstract.

I agree that line 33 is confusing—it could imply that DBS were made from the venous blood of some of the patients and from fingerstick from others, or it could imply that DBS were not made from the blood of all patients included, but the earlier sentence says that DBS were collected from all 1312.

Suggested word changes starting at line 33: “…these patients. Agreement between the Aptima…”. I would suggest dropping the word total throughout because it does not add further understanding of what was analyzed. However, it was used in the previous publication, so keeping it would be consistent.

The authors should mention their previous study in the introduction (not just as a citation), that the current study is an extension of that one with more data, and especially to note that fingerstick and venous DBS were compared in the previous study so they didn’t need to do that here. In the discussion (lines 326-327), they should acknowledge their previous study is their own and that the current study extends from the previous one.

The authors need to add the specific number of samples used in this study in the methods section. This includes the number people with both plasma and DBS, the number with only plasma, and the number with only DBS.

Line 283: change wording to “HIV viral load >1000 copies/mL in Aptima but <1000 copies/mL in Abbott RT”

Lines 305-324: This paragraph should mostly move to the results section because it describes a new comparison of the data. In the Discussion, there should be a new paragraph about this analysis to put the results into context.

Reviewers' comments:

Reviewer's Responses to Questions

**Comments to the Author**

1. If the authors have adequately addressed your comments raised in a previous round of review and you feel that this manuscript is now acceptable for publication, you may indicate that here to bypass the “Comments to the Author” section, enter your conflict of interest statement in the “Confidential to Editor” section, and submit your "Accept" recommendation.

Reviewer #1: (No Response)

2. Is the manuscript technically sound, and do the data support the conclusions?

Reviewer #1: Partly

3. Has the statistical analysis been performed appropriately and rigorously? 

Reviewer #1: Yes

4. Have the authors made all data underlying the findings in their manuscript fully available?

Reviewer #1: No

5. Is the manuscript presented in an intelligible fashion and written in standard English?

Reviewer #1: Yes

6. Review Comments to the Author

Reviewer #1: Review of responses to reviewers’ comments

he major issues to address are the inconsistent numbers throughout, the lack of replicate testing, and

the higher rate of quantitative values from DBS than from plasma. These and other issues are detailed

below:

a. Corrected the inconsistent in N from 439 to 436 for quantified plasma samples throughout the

document

Figure 1 legend still states 439 samples

c. The number of quantifiable results in plasma and DBS sample sets are different because not all

plasma and DBS were prepared from the same patients. Blood samples from only a subset of

patients were used to prepare plasma and DBS for testing in the two assays. The study also

included some DBS samples that did not have corresponding plasma samples and vice versa.

Line 32 of the revised manuscripts states that “Both plasma and DBS samples were collected from each of 1312 adults who participated.” but line 35 DBS was prepared using venous blood from a subset of these patient.” which is conflicting.

Make sure everything is the same between the text and the abstract and the figures (436 vs 439; r = 0.96

vs R = 0.92—need to check both the number and the capitalization for that one; r = 0.87 vs R = 0.8692

need to check number of significant digits and capitalization; mean bias for DBS storage = -0.154 vs -

0.15). Be consistent with your significant digits and with small case r throughout.

Changed correlation coefficient from “R” to “r” in the document

Still capitalized in line 231 of the revised manuscript, R of 0.92 is different to the R= 0.9589 on the figure

Line 184: discordance rates not shown in Table 1 but can be obtained with the positive agreement

values in Table 1, so re-write this statement. This paragraph (lines 184-191) is difficult to understand in

terms of useful results. It is useful to the reader to know how many of the discordant results were due

to higher detection in one assay vs the other, but the value of the rest of the “results” is less clear. What

is the goal of reporting that 17 of the discordant samples (Abbott higher than Aptima in DBS) had

corresponding plasma VL that were higher than 3631cp/mL? Is that VL cutoff a magic number for these

assays? There is no mention of these “results” in the discussion, providing no context for why this

“result” should be noted.

Removed the reference to Table 1 from the statement of discordant rate.

The purpose of pointing out that 31 samples with discordant DBS results at MDP of 1000 c/mL had

plasma results <3631 c/mL is to demonstrate that these samples may flip above and below a 1000 c/mL in different replicates because the HIV concentration is close to a 1000 c/mL. Also added a couple of sentences under discussion to explain the DBS discordants to link with the results in this section.

There is still no indication of why 3631 cp/mL (3,56 log10) specifically was chosen as a cut –off. Using a range of 2,5 – 3,5 log10 incorporating the 0,5 log10 accepted variation in VL assays, would be relevant to justify the flip argument suggested.

Major points

1. It is not clear at abstract’s reading that the authors were working on DBS prepared from venous blood and not from finger-stick. The number of quantifiable samples is different in abstract 436 and results 439.

Added a statement in the methods that venous DBS was used for this study. Corrected the number of quantified samples to 436

Figure 1 still states 439 samples.

6. 439 quantifiable plasma VL but 636 quantifiable fresh VB DBS: this discrepancy is not discussed

The number of quantifiable results in plasma and DBS sample sets are different because plasma and DBS were not prepared from all the same patients. Blood samples from only a subset of patients were used to prepare plasma and DBS for testing in the two assays. The study also included some DBS samples that did not have corresponding plasma samples and vice versa. Statements to clarify this has been added to the methods section.

Does this explanation provided by the authors indicate that the discordant DBS samples were only found in the subgroup that had both plasma and DBS samples types? There are 197 more quantifiable DBS results than plasma which is unusual in a comparison of these sample types particularly since the DBS samples were made from the 1312 enrolled patients and has a higher limit of quantification than plasma. Is this related to the run failure? It is important to clarify the reason for the difference.

Minor points

3. In the methods, a flow chat of tested samples is lacking

A flow chart was not added because only the blood from a subset of the 1312 patients in the plasma

method comparison study were used to prepare DBS. The DBS study also had some patients who did not have plasma results. Statements to clarify this has been added to the methods section

It would still have been easier to understand the sample numbers of the enrolled samples had been described more clearly in the text or in the flow chart suggested above; according the total number of samples enrolled, number of plasma only, both plasma and DBS, and DBS only. The sample numbers are raised several times in the review because it not defined clearly enough in the methods and results.

Second review

Line 24 What criteria were used to define high throughput?

Line 112 Stating that some DBS samples did not have paired plasma samples under the study design heading suggests that was intentional. Is this correct?

Lines 223 – 225 What happened to the other 10 upward misclassified DBS samples?

Line 321 – Does this include the Aptima plasma? it is not clear.

Line 284 – No discussion about possible under quantification but the Realtime assay.

A general review of the grammar and formatting is required including spacing, punctuation and duplication of a few words.

7. PLOS authors have the option to publish the peer review history of their article (what does this mean?). If published, this will include your full peer review and any attached files.

Reviewer #1: No

---

## [Author Response · Author response to Decision Letter 1]

6 May 2022

Rebuttal notes

Line 30: remove the word consenting—this is not needed in the abstract.

The word “consenting” has been removed from line 30.

I agree that line 33 is confusing—it could imply that DBS were made from the venous blood of some of the patients and from fingerstick from others, or it could imply that DBS were not made from the blood of all patients included, but the earlier sentence says that DBS were collected from all 1312.

We have rewritten the methods section (lines 27-34) to read as follows:

This was a cross-sectional study of 2227 HIV infected adults visiting health facilities in Nairobi and Busia, Kenya. Each provided a venous blood sample; plasma was prepared from 1312 samples while paired DBS samples and plasma were prepared from the remaining 915 samples. The agreement between the Aptima assay and the Abbott RealTime HIV-1 Assay (Abbott RT) was analysed by comparing the HIV-1 viral load in both assays at the medical decision point of 1000 copies/mL. To assess stability of HIV-1 RNA in DBS, viral load in DBS spotted on day 0 were compared with that from the same DBS card after 21 days of storage at room temperature.

Suggested word changes starting at line 33: “…these patients. Agreement between the Aptima…”. I would suggest dropping the word total throughout because it does not add further understanding of what was analyzed. However, it was used in the previous publication, so keeping it would be consistent.

We have deleted several “total” words throughout the manuscript. We have left a few of the words where the sense would be lost if we deleted them.

The authors should mention their previous study in the introduction (not just as a citation), that the current study is an extension of that one with more data, and especially to note that fingerstick and venous DBS were compared in the previous study so they didn’t need to do that here. In the discussion (lines 326-327), they should acknowledge their previous study is their own and that the current study extends from the previous one.

We have rewritten the paragraph that starts from line 85 to capture these details as follows:

In a previous study, we investigated the performance of the Aptima HIV Quant Dx assay using fingerstick and venous dried blood spots prepared under field conditions, although we did not compare HIV-1 VL in Aptima with those in other assays using DBS. In fact, whereas multiple studies compared viral load results of paired DBS and plasma samples at the medical decision point of 1000 copies/ mL by testing both sample types in Aptima [21-23, 31], there is no published information comparing the performance of DBS specimens tested in Aptima and other assays at the medical decision point (MDP) of 1000 copies/mL. 

The authors need to add the specific number of samples used in this study in the methods section. This includes the number people with both plasma and DBS, the number with only plasma, and the number with only DBS.

We have rewritten the methods section to make sure this is very clear, as follows:

Whole blood was collected from 2227 participants by phlebotomy, shipped to KEMRI HIV laboratories within 6 hours of collection, and processed within 12 hours of receipt. A total of 1312 samples provided only plasma. Paired DBS and plasma samples were prepared from a further 915 venous samples. To prepare plasma, the whole blood samples were centrifuged at 1,100g for 10 minutes within 24 hours of collection, and the plasma was stored at -800C. To prepare DBS, 70µl of venous blood was spotted in each of ten spots (2 DBS cards) per patient. The DBS samples were allowed to dry overnight. The first DBS card for each patient was used for day 0 testing, while the DBS cards for the 21-day time point were packaged with desiccants and stored at room temperature. All samples were de-identified prior to Aptima testing. 

Line 283: change wording to “HIV viral load >1000 copies/mL in Aptima but <1000 copies/mL in Abbott RT”

The word “but” has been added, apologies for the oversight in this sentence.

Lines 305-324: This paragraph should mostly move to the results section because it describes a new comparison of the data. In the Discussion, there should be a new paragraph about this analysis to put the results into context.

The results had already been presented in the results section, and for this reason we have carefully revised the paragraph to lighten it while still retaining its meaning, from lines 317-329

Reviewer #1: Review of responses to reviewers’ comments

The major issues to address are the inconsistent numbers throughout, the lack of replicate testing, and the higher rate of quantitative values from DBS than from plasma. These and other issues are detailed below:

a. Corrected the inconsistent in N from 439 to 436 for quantified plasma samples throughout the document:

n is now corrected to 436 where appropriate

Figure 1 legend still states 439 samples. 

This has been corrected. Line 234 now reads: Figure 1a: HIV viral load comparison for 436 plasma samples tested in Aptima and Abbott RT assays 

c. The number of quantifiable results in plasma and DBS sample sets are different because not all plasma and DBS were prepared from the same patients. Blood samples from only a subset of patients were used to prepare plasma and DBS for testing in the two assays. The study also included some DBS samples that did not have corresponding plasma samples and vice versa.

In the methods section, we have clarified the exact number of samples of each type used in the study.

Please see lines 127-136

Line 32 of the revised manuscripts states that “Both plasma and DBS samples were collected from each of 1312 adults who participated.” but line 35 DBS was prepared using venous blood from a subset of these patient.” which is conflicting.

This is our error. We have rewritten lines 27 to 31 to read as follows:

This was a cross-sectional study of 2227 HIV infected adults visiting health facilities in Nairobi and Busia, Kenya. Each provided a venous blood sample; plasma was prepared from 1312 samples. Paired DBS and plasma samples were prepared from the remaining 915 samples. 

Make sure everything is the same between the text and the abstract and the figures (436 vs 439; r = 0.96

vs R = 0.92—need to check both the number and the capitalization for that one; r = 0.87 vs R = 0.8692

need to check number of significant digits and capitalization; mean bias for DBS storage = -0.154 vs -

0.15). Be consistent with your significant digits and with small case r throughout.

Changed correlation coefficient from “R” to “r” in the document

This has now been done throughout the document.

Still capitalized in line 231 of the revised manuscript, R of 0.92 is different to the R= 0.9589 on the figure.

We have corrected this error in the manuscript, and resolved capitalisation around r.

Line 184: discordance rates not shown in Table 1 but can be obtained with the positive agreement values in Table 1, so re-write this statement. This paragraph (lines 184-191) is difficult to understand in terms of useful results. It is useful to the reader to know how many of the discordant results were due to higher detection in one assay vs the other, but the value of the rest of the “results” is less clear. What is the goal of reporting that 17 of the discordant samples (Abbott higher than Aptima in DBS) had corresponding plasma VL that were higher than 3631cp/mL? Is that VL cutoff a magic number for these assays? There is no mention of these “results” in the discussion, providing no context for why this “result” should be noted.

We have removed the reference to Table 1 from the statement of discordant rates.

The purpose of pointing out that 31 samples with discordant DBS results at MDP of 1000 c/mL had plasma results <3631 c/mL is to demonstrate that these samples may flip above and below a 1000 c/mL in different replicates because the HIV concentration is close to a 1000 c/mL. We have also added a couple of sentences under the discussion to explain the DBS discordants to link with the results in this section.

There is still no indication of why 3631 cp/mL (3,56 log10) specifically was chosen as a cut –off. Using a range of 2,5 – 3,5 log10 incorporating the 0,5 log10 accepted variation in VL assays, would be relevant to justify the flip argument suggested.

Your point is well noted. We agree that there is nothing special about the 3631cp/ml. We have rewritten the sentence (lines 326-329) as follows: 

The remaining 14 of the 31 patients with Aptima DBS VL <3 log c/mL and Abbott DBS VL>3 log c/mL had plasma VL >3.00 log copies/mL in both Aptima plasma and Abbott plasma suggesting that these were upward misclassifications by Abbott DBS.

Major points

1. It is not clear at abstract’s reading that the authors were working on DBS prepared from venous blood and not from finger-stick. The number of quantifiable samples is different in abstract 436 and results 439.

We have added a statement in the methods that venous DBS was used for this study. Please see line 23.

We have corrected the number of quantified samples to 436, and indicted that venous DBS was used, throughout the document.

Figure 1 still states 439 samples.

We have corrected this to 436 samples in Fig. 1.

6. 439 quantifiable plasma VL but 636 quantifiable fresh VB DBS: this discrepancy is not discussed.

The 436 quantifiable plasma samples were prepared from venous blood taken from 1312 patients. The 636 quantifiable DBS samples were prepared from a further group of (915) participants, who also provided plasma. 

For that reason the yields are not comparable.

Does this explanation provided by the authors indicate that the discordant DBS samples were only found in the subgroup that had both plasma and DBS samples types? 

As clarified, there were 915 paired DBS and plasma samples. The 49 discordant DBS samples are a subset of the 915 samples.

There are 197 more quantifiable DBS results than plasma which is unusual in a comparison of these sample types particularly since the DBS samples were made from the 1312 enrolled patients and has a higher limit of quantification than plasma. Is this related to the run failure? It is important to clarify the reason for the difference.

These results arise from different sets of participants. So that comparison does not arise. Please see lines 127-136.

Whole blood was collected from 2227 participants by phlebotomy, shipped to KEMRI HIV laboratories within 6 hours of collection, and processed within 12 hours of receipt. A total of 1312 samples provided only plasma. Paired DBS and plasma samples were prepared from a further 915 venous samples.

The 49 discordant DBS samples were part of the 915 paired DBS and plasma samples tested.

Minor points

3. In the methods, a flow chat of tested samples is lacking.

A total of 1312 samples provided only plasma. Paired DBS and plasma samples were prepared from a further 915 venous samples. Since there was no further dichotomy, a flow chart was not necessary.

It would still have been easier to understand the sample numbers of the enrolled samples had been described more clearly in the text or in the flow chart suggested above; according the total number of samples enrolled, number of plasma only, both plasma and DBS, and DBS only. The sample numbers are raised several times in the review because it not defined clearly enough in the methods and results.

We have extensively rewritten this section to make it as clear as possible. Please see lines 179-185.

Second review

Line 24 What criteria were used to define high throughput?

There are no universally agreed criteria for defining high throughput. In our setting, we consider low throughput to mean <96 tests in 24 hours, medium throughput to mean 97-480 tests in 24 hours and high throughput to be >481 tests in 24 hours.

Line 112 Stating that some DBS samples did not have paired plasma samples under the study design heading suggests that was intentional. Is this correct?

This was our mistake. (In actual fact, this error occurred out of confusion from: 258 of our DBS samples had accompanying fingerstick samples while the rest did not, but that data is reported elsewhere already). 

As stated in the manuscript, a total of 1312 samples provided only plasma. Both plasma and paired DBS samples were prepared from a further 915 venous samples.

Lines 223 – 225 What happened to the other 10 upward misclassified DBS samples?

We have revised this section so that the discussion does not mislead, from lines 335-346.

Line 321 – Does this include the Aptima plasma? it is not clear.

The upward misclassification was seen only in Aptima DBS, not Aptima or Abbott plasma or Abbott DBS.

Line 284 – No discussion about possible under quantification by the Realtime assay.

The discussion around this finding has now been provided, lines 331-342

A general review of the grammar and formatting is required including spacing, punctuation and duplication of a few words.

We have used the grammar tool to resolve spacing challenges and correct grammatical errors.

---

## [Editor Report · Decision Letter 2]

30 May 2022

Comparison of the performance of Aptima HIV-1 Quant Dx Assay with Abbott RealTime HIV Assay for viral load monitoring using plasma and Dried Blood Spots collected in Kenya

PONE-D-21-27672R2

Dear Dr. Mwau,

We’re pleased to inform you that your manuscript has been judged scientifically suitable for publication and will be formally accepted for publication once it meets all outstanding technical requirements.

Kind regards,

Julie AE Nelson, PhD

Academic Editor

PLOS ONE

Additional Editor Comments (optional):

Changes to make before publication:

Line 51: Choose one of these options: add “(VL)” after “viral load” here and change to VL for all subsequent uses, OR change all uses of “VL” to “viral load”. Do not use them interchangeable throughout.

Line 66: add “, respectively” after “platforms”

Line 76: add comma before “respectively”

Line 130: remove “; of those”

Line 132: fix “-800C” to “-80°C”

Line 139: add a short description and citation for the DBS testing on the Abbott system since it is not described in the package insert. A description should include how much of a spot used and how much volume of which buffer was used for elution. It would also be nice for the reader to include a similar description for the Aptima assay at line 142 so readers don’t need to go to the package insert. There is a description of how many spots were used on lines 155-156—move this information up to line 139-143.

Line 159: Specify whether the conversion factor done by the tech or by the software on the Aptima. What about the conversion factor for Abbott?

Line 190: the 2.94 copies/mL and 7 copies/mL should say log copies/mL instead. However, please convert these numbers to copies/mL for the Aptima to be equivalent to the Abbott description for better readability.

Line 230: You have said that 1312 plasma samples were compared in the assays and that 915 DBS samples were compared in the assays. In this sentence, however, you now indicate that the plasma samples were tested that correspond to the DBS samples. Is there a reason that you did not include these 915 plasma samples results with the 1312 results? Also, if both plasma and DBS were made for 915 patients, were both plasma and DBS tested on Abbott for delivery of patient results? It is ok to not include the plasma results with the 1312, but it would be helpful to the reader to explain why they were not reported that way.

Line 296: Remove the phrase about the 0.33 log higher quantification, stopping the sentence after the first “Abbott RT” in this line. The later discussion of the 0.33 log is more clear and this one is just confusing.

Line 324: This sentence got jumbled. Remove the “>3 log c/mL in both Aptima plasma and Abbott plasma”. Here I also notice that you are interchangeable using log c/mL and log copies/mL. Use just one way throughout the manuscript.

Line 344: rewrite as follows: “Although only venous DBS were tested in this study, we previously compared the recovery of HIV-1 from venous DBS, fingerstick DBS, and plasma with the Aptima test [31], showing good agreement between plasma, venous DBS, and fingerstick DBS.”

Line 346: change “this” to “the current”
---

## [Editor Report · Acceptance letter]

8 Jun 2022

PONE-D-21-27672R2 

Comparison of the performance of Aptima HIV-1 Quant Dx Assay with Abbott RealTime HIV Assay for viral load monitoring using plasma and Dried Blood Spots collected in Kenya 

Dear Dr. Mwau:

I'm pleased to inform you that your manuscript has been deemed suitable for publication in PLOS ONE. Congratulations! Your manuscript is now with our production department. 

Kind regards, 

on behalf of

Dr. Julie AE Nelson 

Academic Editor

PLOS ONE